# Three Years of COVID-19 on Orthopaedic Trauma; Are We Going Back to Normality?

**DOI:** 10.3390/medicina59081464

**Published:** 2023-08-16

**Authors:** Vittorio Candela, Riccardo Proietti, Giuseppe Polizzotti, Marco Rionero, Stefano Gumina

**Affiliations:** 1Department of Anatomical, Histological, Forensic Medicine and Orthopaedics Sciences, Sapienza University of Rome, 00185 Rome, Italy; riccardoproiet@libero.it (R.P.); giuseppe.polizzotti@uniroma1.it (G.P.); marcorionero@gmail.com (M.R.); stefano.gumina@uniorma1.it (S.G.); 2Istituto Chirurgico Ortopedico traumatologico (ICOT), 04100 Latina, Italy

**Keywords:** COVID-19 pandemic, COVID-19 orthopaedic, COVID-19 traumatology, COVID-19 impact on trauma surgery, orthopaedic trauma during COVID-19, COVID-19 impact on shoulder trauma, COVID-19 impact on elbow trauma

## Abstract

*Background and Objective:* On March 2020, our country became a protected area due to the COVID-19 pandemic. The consequences of COVID-19 on trauma surgery were great. We aimed to evaluate the activity of the Trauma Centre of a highly populated suburban area over 30 days starting from the first day of restrictions, to compare it with the same period of 2019 and 2022 and to evaluate whether a progressive return to normality has taken place. *Materials and Methods*: All patients older than 18 years managed in our Trauma Unit between 8 March 2020 and 8 April 2020 (the first COVID-19 period) were compared to the same period of 2019 (a COVID-19 free period) and 2022 (the second COVID-19 period). Clinical records were examined. Five categories of diagnoses and six mechanisms of injury were distinguished. *Results*: There were 1351 patients [M:719–F:632; mean age (SD):49.9 (18.7)], 451 [M:228–F:223; mean age (SD):55.9 (18.4)] and 894 [M:423–F:471;mean age (SD):54.1 (16.7)] in the COVID-19 free and in the first and second COVID-19 periods, respectively (*p* < 0.05). In 2020, the most significant decrease was registered for sprains/subluxations (80%); contusions decrease by 77% while fractures decrease only by 37%. The lowest reduction was found for dislocations (26%). In 2022, dislocations decreased by only 16% and both fractures and sprains decreased by about 30% with respect to the pre-pandemic period. Patients with minor trauma (contusions) were half compared to 2019. Accidental falls remain the most frequent mechanism of injury. The incidence of proximal femur, proximal humerus and distal radius fractures remained almost unchanged during both pre-pandemic and pandemic periods. *Conclusions*: COVID-19 has markedly altered orthopaedic trauma. Injuries related to sports and high energy trauma/traffic accidents drastically reduced in 2020; however, we are slowly going back to normality: the same injuries increased in 2022 due to the progressive easing of restrictions. Elderly fractures related to accidental falls remained unchanged.

## 1. Introduction

On 29 January 2020 the first cases of Coronavirus were declared in Italy corresponding to two Chinese tourists coming from China.

On 30 January 2020, a global state of emergency was set; on 11 February 2020, the new disease was named COVID-19 by the World Health Organization. On 21 February 2020, several cases of COVID-19 were diagnosed all around Italy. 

On 4 March 2020, schools and universities were closed; there were more than 2500 positive cases.

On 8 March 2020, Italy became a “protected area”. People across Italy were allowed to leave their homes only for an essential reason. Smart working was mandatory and movements between states and towns were forbidden. Travel was only allowed for “urgent, verifiable work situations and emergencies or health problems”. Inhabitants were allowed to go outside for a work-related and health reasons and to buy food.

These restrictions have been in place for three months; during the summer, social spacing was the sole rule with a consequent progressive raise of the infection at the end of September 2020. Again, the Italian Government has progressively limited the movements between regions, imposed smart working for all services not considered as essential, and prohibited sports activities in gyms and all team sports. Furthermore, a curfew from 10 PM to 7 AM was established.

In 2021, the vaccination campaign started, hospital bed occupancy was under control and a climate of cautious optimism began. 

COVID-19 has inevitably modified everyday life for three years. The effects have also been evident in orthopaedic trauma [1,2,3,4,5,6,7,8,9,10,11,12,13]. The incidence of fractures significantly decreased all around the world, with the exception of some joints such as the femoral neck [1,3,4,5,10,11,13,14,15,16,17,18,19,20,21,22,23,24,25,26,27,28,29,30,31]. However, information is relative to the first year of the COVID-19 pandemic and no data are present regarding a definitive return to the pre-COVID period. 

Our aim was to assess the activity of the Trauma Centre of a large highly populated suburban area over a 30-day period starting from 8 March 2020, the first day of lockdown in Italy, and comparing it with the same days of 2019 and to a 30-day period of the 2022 lockdown period in order to both weigh the impact of COVID-19 on orthopaedic trauma and to say whether there has been a “definitive end”. 

## 2. Materials and Methods

All admitted patients of the Trauma Unit of our hospital between 8 March 2020 and 8 April 2020 (first COVID-19 period) were retrospectively included and compared to patients admitted in the same period of 2019 (COVID-19 free period) and of 2022 (second COVID-19 period). Patients < 18 years of age were excluded.

Information regarding age, sex, involved side, mechanism of injury and diagnosis was collected by two of the authors.

Five different types of diagnosis were detected: fracture, contusion, sprain/subluxation, dislocation and “other”. Other included wounds; muscle injuries; tendon lesions; and articular pain without trauma.

Six subgroups of mechanisms of injury were arbitrarily distinguished: (1) accidental fall at home/in the street; (2) sports trauma; (3) high-energy trauma that occurred from car, motorcycle, public transport or pedestrian involvement; (4) accident at work; (5) trauma due to assaults or beatings; and (6) no trauma.

## 3. Statistical Analysis

Continuous variables were expressed by the mean and standard deviation (SD) and were evaluated by the Student’s *t*-test or Mann–Whitney U test. The categorical data were expressed as a number and percentage (%) and were evaluated by the chi-square or Fisher’s exact test. The statistical test level was set as *p* < 0.05. SPSS23.0 was used to perform all the tests (IBM, Armonk, NY, USA).

## 4. Results

There were 1351 patients admitted to the Trauma Unit [M:719–F:632, mean age (SD): 49.9 (18.7)], 451 [M:228–F:223, mean age (SD):55.9 (18.4)] and 894 [M:423–F:471, mean age (SD): 54.1 (16.7)] in the COVID-19 free and in the first and second COVID-19 periods, respectively (*p* < 0.05). 

The distribution of injuries in the three periods according to the five diagnostic groups are reported in Figure 1. A significant difference was found between the COVID-free period and the first COVID period (*p* < 0.01) and between the first and second COVID periods (*p* < 0.02) according to all diagnostic groups. 

Figure 2 shows the distribution of fractures in the three studied periods. A significant difference was found between the COVID-free period and the first COVID period according to shoulder, elbow and hand fractures (*p* < 0.01) and between the first and second COVID periods according to shoulder and elbow fractures (*p* < 0.03).

Relative to joint dislocations, the prevalence of shoulder dislocations remains constant during the studied period, going from 53% (COVID-free period) to 50% and 56% (first and second COVID periods, respectively). Only a slight decrease was registered for hip prosthesis dislocations (from 10% to 7%).

The prevalence of ankle sprains was 58% in the COVID-free period, relative to all joint sprains. During the first COVID period, the absolute number of ankle sprains significantly decreased (from 137 to 33) and returned almost back to normality in the 2022 period (absolute number 109).

According to the mechanism of injury, Figure 3 shows the different distributions of the five diagnostic groups. 

## 5. Discussion

It is the first study with the specific aim of evaluating the effectiveness of a possible return to normality in the orthopaedic practice comparing both pre- and post-COVID periods. We registered a 67% reduction in services provided at our Trauma Centre during the first pandemic period. These data are similar to the findings of Migliorini et al. [32] who recently published a brilliant analysis of 57 clinical investigations concluding that the overall reduction in surgical interventions ranged from 5.4 to 88.8% [32,33].

In 2022, we recorded only decrease of a third (34%) compared to the pre-pandemic period. These data confirm the feeling of optimism linked to the implementation of the vaccination campaign. Focusing on the diagnosis, significant differences emerge regarding the first COVID-19 period: the greatest decrease in the first COVID-19 period was registered for sprains/subluxation (80%); contusions decreased by 78% while fractures decreased by only 37%. The lowest decrease was found for dislocations (26%). In the second COVID-19 period (2022), dislocations decreased by only 16% and both fractures and sprains decreased by about 30% with respect to the pre-pandemic period. On the other hand, the number of patients with minor trauma (contusions) was halved compared to 2019.

It is plausible that two main causes led to this dramatic decrease when comparing 2019 and 2020. The first is purely “medical”: The restrictions imposed to reduce the COVID-19 spread have inevitably cut down the circumstances that usually predispose to trauma: sports injuries and, to a lesser extent, accidents at work. The decrease in the number of the ankle (76%) and knee (92%) sprains in 2020 confirms this hypothesis. The second one is “social”: Restrictions, together with the fear of possible contagion, discouraged those who suffered minor trauma or low-intensity pain from going to our Trauma Unit. On the contrary, the same patients did not hesitate to go to the Trauma Unit in the COVID-19 free period thanks to the free admission of the national health system. In 2021, the partial reopening of both the sports and working activities, led to an inevitable increase in access and these data reflect the gradual increase in the Trauma Unit access in the 2022. 

Analysing the mechanisms of injury, a significant decrease in sports trauma and accidents at work was present in 2020; again, in 2022, these mechanisms of injury were increasing. On the other hand, a relative persistence of high-energy road accidents was found in both 2020 and 2022. Our analysis was carried out in a highly populated suburban area with a high flow and hazardous roads; it is probable that the restrictive measures led to a decrease in the circulating cars/motorcycles with a simultaneous increase in the running speed of those who were outside for one of the reasons provided by the law.

Accidental falls remained the most frequent mechanism of injury and proximal femur, proximal humerus and distal radius were the most frequent site of fractures in both pre-pandemic and pandemic periods; these fractures were only minimally affected by COVID-19 (decreased by 12% for the proximal femur, 16% for the proximal humerus and 19% for the distal radius in 2020; decreased by only 3% for the proximal femur, 7% for the proximal humerus and 15% for the distal radius in 2022). These injuries are the consequence of low-energy injury-at-home trauma in elderly osteoporotic patients. Once again, these results, although they go beyond the main aim of the present study, highlighted the importance of home prevention measures.

## 6. Conclusions

COVID-19 has markedly altered orthopaedic trauma. Injuries related to sports and high-energy trauma or traffic accidents drastically reduced in 2020; however, our study demonstrated that we are going back to normality: the same injuries increased in 2022 due to the easing of restrictions and the return to normal life.

The rate of elderly osteoporotic fractures related to accidental falls remained unchanged with respect to the pre-pandemic period. The present study highlights the importance of primary prevention measures, especially at home, and could be used as a reference for individuals, health care providers and health administrative departments.

## Figures and Tables

**Figure 1 medicina-59-01464-f001:**
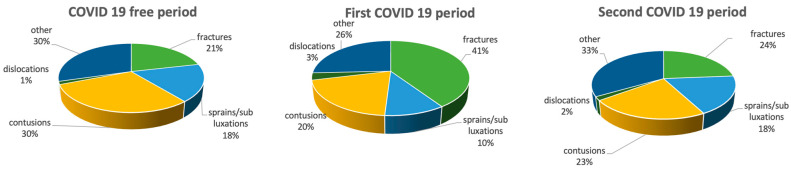
Distribution of injuries in the three different periods according to the five diagnoses. “Other” includes wounds, muscle injuries, tendon lesions and articular pain without trauma.

**Figure 2 medicina-59-01464-f002:**
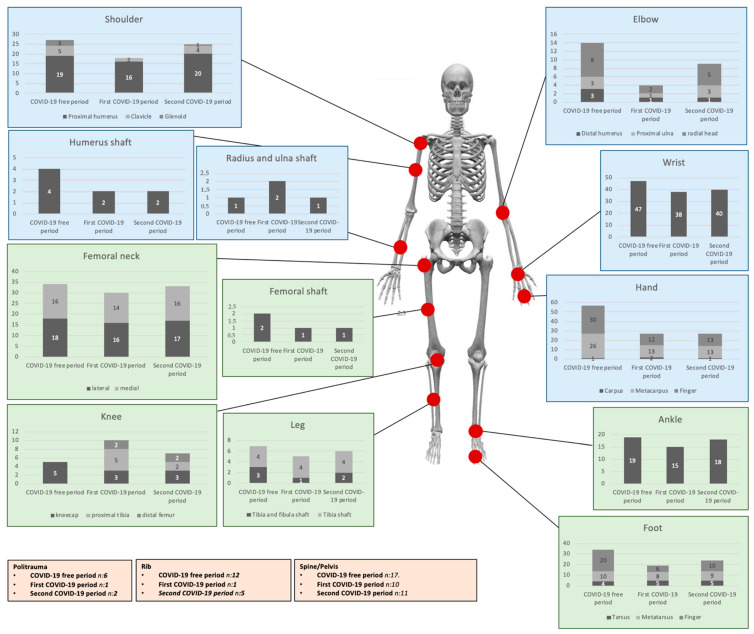
Distribution of fractures in the COVID-19 free and COVID-19 periods.

**Figure 3 medicina-59-01464-f003:**
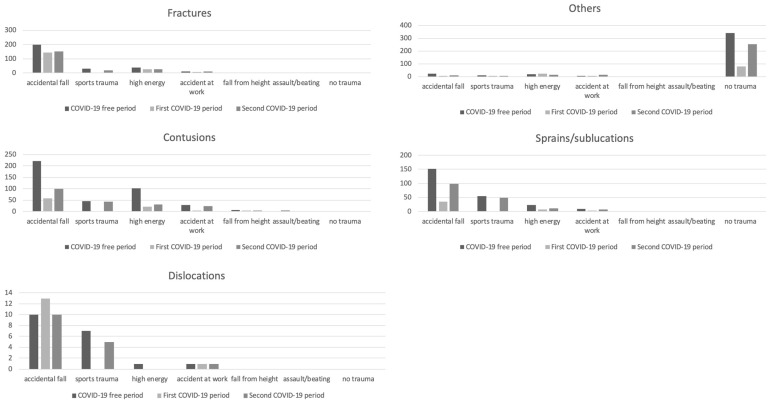
Mechanisms of injury responsible for fractures, contusions, dislocations, sprains/subluxations and “other” in the two studied periods.

## Data Availability

Not applicable.

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
