# Peer review of "Three Years of COVID-19 on Orthopaedic Trauma; Are We Going Back to Normality?"

_medicina, 2023, doi:10.3390/medicina59081464_

Round 1

Reviewer 1 Report

Following are my observations:

1.     The number of pie charts may be decreased. A table comparing the incidence of various injuries in different time periods may be better for comparison and comprehension.  

2.     It is understandable that injuries will decrease once the activities predisposing to risk of trauma are curtailed.

3.     Interestingly, many injuries have increased if 1st and 2nd COVID-19 period is compared to each other. The authors should mention the reason or their hypothesis for same.  

Author Response

REVIEWER 1

The number of pie charts may be decreased. A table comparing the incidence of various injuries in different time periods may be better for comparison and comprehension.  

-Figures were decrease and now information is more direct.

Interestingly, many injuries have increased if 1st and 2nd COVID-19 period is compared to each other. The authors should mention the reason or their hypothesis for same.  

-Discussion was modified as suggested.

Reviewer 2 Report

The authors made an analysis of Three year of COVID-19 on the orthopaedic trauma. The reseah is interesting, however they have to make some modifications to the manuscript.

The figures I, 3 5 etc, could be more understandable if they are presented in a single figure (as columns chart type), besides it save space on the manuscript and the reader could see the behavior of every trauma

 Finally, I suggest to author, clarify how this study helps to prevent traumas or how this investigation will be useful in trauma or medicine area?

Author Response

The figures I, 3 5 etc, could be more understandable if they are presented in a single figure (as columns chart type), besides it save space on the manuscript and the reader could see the behavior of every trauma

-Figures 1 and 7 were modified; Figure 3-6 were deleted and data were added in the result section;

 Finally, I suggest to author, clarify how this study helps to prevent traumas or how this investigation will be useful in trauma or medicine area?

Conclusion was modified: It is important to underline the fact the femoral neck fracture have remained an issue during all the pandemic period

Reviewer 3 Report

Reading the introduction, you get the feeling that the authors wrote the paper a year ago. "These rules are still in force nowadays due to virus variants appearance. However, the vaccination campaign is started, the hospital beds occupancy is under control and a climate of cautious optimism is beginning to breathe throughout the country." According to my information, no restrictions have been in force since July 2022.

The pie charts in Figure 3 are poorly suited for presenting the data and do not allow for comparison of the data with each other. Here, a table would be better to present the data.

Figure 4 shows in three diagrams that there were no differences in the years and is thus much too large for the content.

The results section consists of a few sentences and no statistics. The statistical analysis with SPSS, which is mentioned in Material and Methods, including the tests mentioned, does not take place. 

The first 13 lines of the discussion are a repetition of the introduction and unnecessary or wrong at this point.

In the last two lines on page 13, the text is in Italian - should it still be translated? This should not be overlooked in the proofreading!

The entire discussion only describes the data of the article and is therefore not a scientific discussion of the topic. Much of the discussion is actually results. Comparison to other studies, discussion of similar work etc is completely missing.

Significant differences are mentioned, but the data in the Results section is neither described nor analysed. 

Is there an ethics vote for this investigation or is such a vote not necessary in Italy? My local insights are not sufficient here.

In the last two lines on page 13, the text is in Italian - should it still be translated? This should not be overlooked in the proofreading!

Author Response

REVIEWER 3

Reading the introduction, you get the feeling that the authors wrote the paper a year ago. "These rules are still in force nowadays due to virus variants appearance. However, the vaccination campaign is started, the hospital beds occupancy is under control and a climate of cautious optimism is beginning to breathe throughout the country." According to my information, no restrictions have been in force since July 2022.

The introduction was rephrased; in our Country few restrictions are still in force in 2023.

The pie charts in Figure 3 are poorly suited for presenting the data and do not allow for comparison of the data with each other. Here, a table would be better to present the data.

Figure 4 shows in three diagrams that there were no differences in the years and is thus much too large for the content.

Figures 1 and 7 were modified; Figure 3-6 were deleted and data were added in the result section;

The results section consists of a few sentences and no statistics. The statistical analysis with SPSS, which is mentioned in Material and Methods, including the tests mentioned, does not take place. 

Statistics was added in the result section.

The first 13 lines of the discussion are a repetition of the introduction and unnecessary or wrong at this point.

Deleted as suggested.

In the last two lines on page 13, the text is in Italian - should it still be translated? This should not be overlooked in the proofreading!

Modified as suggested.

The entire discussion only describes the data of the article and is therefore not a scientific discussion of the topic. Much of the discussion is actually results. Comparison to other studies, discussion of similar work etc is completely missing.

Discussion was modified as suggested.

Round 2

Reviewer 3 Report

Thank your for your renewed manuscript. I think it improved a lot.

P 4, Line 9: should it be p<0.02 or p>0.02??

Same in the last lines of page 4: "<" or ">"?

Author Response

P 4, Line 9: should it be p<0.02 or p>0.02??

Same in the last lines of page 4: "<" or ">"?

correct as suggested